# Development of a 1,3a,6a-triazapentalene derivative as a compact and thiol-specific fluorescent labeling reagent

Atsushi Nakayama [1,2], Akira Otani[1], Tsubasa Inokuma[1,2], Daisuke Tsuji[1], Haruka Mukaiyama[1], Akira Nakayama [3], Kohji Itoh[1], Akira Otaka[1], Keiji Tanino [4] & Kosuke Namba [1,2]*

For the fluorescence imaging of biologically active small compounds, the development of compact fluorophores that do not perturb bioactivity is required. Here we report a compact derivative of fluorescent 1,3a,6a-triazapentalenes, 2-isobutenylcarbonyl-1,3a,6a-triazapentalene (TAP-VK1), as a fluorescent labeling reagent. The reaction of TAP-VK1 with various aliphatic thiols proceeds smoothly to afford the corresponding 1,4-adducts in high yields, and nucleophiles other than thiols do not react. After the addition of thiol groups in dichloromethane, the emission maximum of TAP-VK1 shifts to a shorter wavelength and the fluorescence intensity is substantially increased. The utility of TAP-VK1 as a compact fluorescent labeling reagent is clearly demonstrated by the labeling of Captopril, which is a small molecular drug for hypertension. The successful imaging of Captopril, one of the most compact drugs, in this study demonstrates the usefulness of compact fluorophores for mechanistic studies.

[1] Department of Pharmaceutical Sciences, Tokushima University, 1-78-1 Shomachi, Tokushima 770-8505, Japan. [2] Research Cluster on "Innovative Chemical Sensing", Tokushima University, 1-78-1 Shomachi, Tokushima 770-8505, Japan. [3] Department of Chemical System Engineering, Graduate School of Engineering, The University of Tokyo, Bunkyo-ku, Tokyo 113-8656, Japan. [4] Department of Chemistry, Faculty of Science, Hokkaido University, Kita-ku, Sapporo 060-0810, Japan. *email: namba@tokushima-u.ac.jp

Fluorescent organic molecules are an important class of compounds in modern science and technology, and they are widely used in biological imaging probes, sensors, lasers, and light-emitting devices[1–3]. The development of useful fluorescent organic molecules is thus crucial for the advancement of many industries, and has been a subject of intensive research[4–6]. The field of chemical biology in particular has recently seen rapid advances due to the development of useful fluorescent probes, and various biological phenomena have thereby been elucidated[7–11]. The introduction of organic fluorophores into biologically active molecules such as drugs, natural products, hormones, peptides, and proteins has been one of the most useful ways to study these molecules mechanistically by visualizing their localization and behavior inside cells and tissues[12–14]. However, a key improvement is still needed for commonly used fluorescent organic molecules. Most highly fluorescent molecules are large in relation to bioactive organic compounds, and fluorescence-labeled compounds sometimes lose their biological activities as a result of the structural modifications. Even proteins that interact with large biomolecules can lose their interaction affinities when fluorescent molecules are introduced near the binding site. Thus, the development of compact fluorescent labeling reagent has still been strongly demanded (Fig. 1).

Herein, we report the successful development of a compact fluorescent labeling reagent that specifically reacts with aliphatic thiols based on the 1,3a,6a-triazapentalene (TAP) skeleton, in which the usefulness of the compact fluorophore is clearly demonstrated by the fluorescence imaging of small-molecule drug in living cells.

## Results

**Development of a most compact derivative of 1,3a,6a-triaza-pentalene.** We recently discovered that a 1,3a,6a-triazapentalene (TAP) (**1**) skeleton is a compact and highly fluorescent chromophore with the potential to overcome the molecular size problem (Fig. 2)[15], and it has been applied to various fluorescent probes[16–19]. Interestingly, the fluorescence of the TAPs shifted to longer wavelengths along with the inductive effect of the 2-substituents[15,19–24]. By exploiting this effect, we recently developed yellow and red fluorescent 1,3a,6a-triazapentalene derivatives that are relatively smaller than the conventional yellow and red fluorescent molecules, as shown in Fig. 2[25]. However, these derivatives might be made still more compact: the phenyl group at the C2-position could be removed to reduce their overall molecular size. We thus develop a most compact fluorescent labeling reagent without 2-phenyl group. First, we investigated the effective functional groups at the 2-position as well as the phenyl group, and we found that the ketone group was the most stable against photoirradiation (Supplementary Note 1 and Supplementary Table 1). To develop the ketone analog as a compact fluorescent labeling reagent, we designed the vinyl ketone **2** as shown in Fig. 2. The vinyl ketone moiety of **2** was expected to readily react with thiol groups contained in various biologically

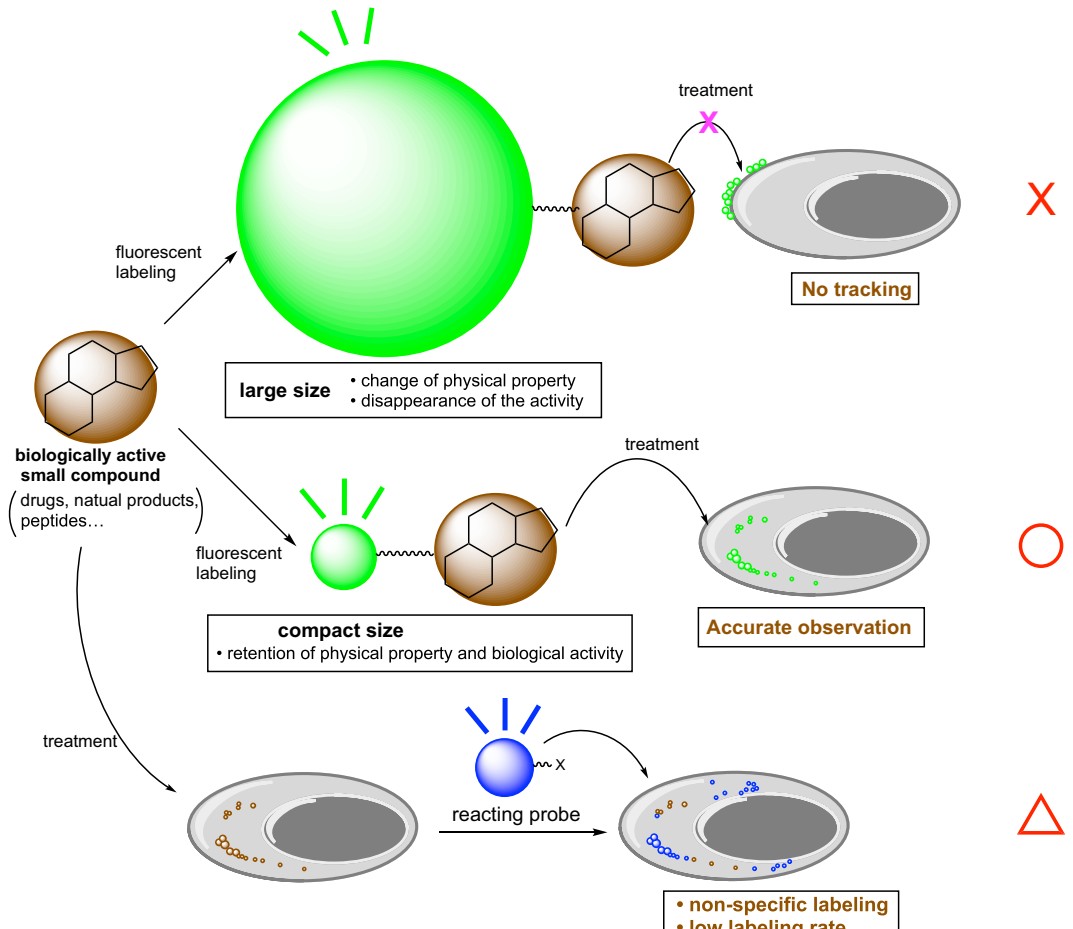

**Fig. 1 Desirable fluorescent labeling group for intracellular tracking of biologically active compounds.** Since the large molecular size of fluorescent groups can diminish the biological activity of the parent small compound, the smallest possible molecular size is demanded. On the other hand, intracellular reacting probes have problems with nonspecific labeling and low labeling rates. X = no good, O = good and Triangle = moderate.

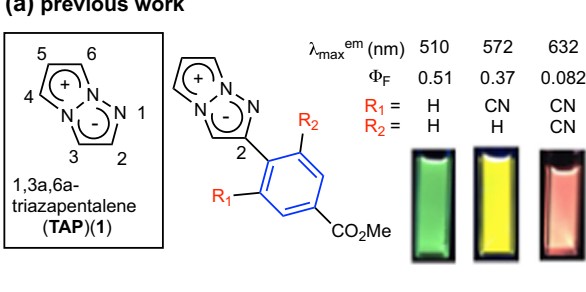

**(a) previous work**

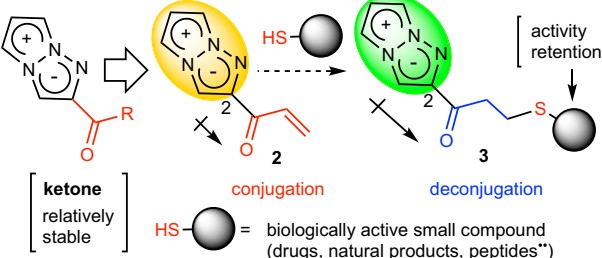

**(b) this work (to be most compact)**

**Fig. 2 Structure and fluorescence properties of 1,3a,6a-triazapentalene and the expansion to the most compact fluorescent labeling reagent.** **a** Previous work; structure of 1,3a,6a-triazapentalene (TAP) and the fluorescence properties of TAP that change according to the inductive effect at the C2-position. **b** This work; the ketone group was observed to be a more photostable functional group than alkyl, amide, or ester. To use the ketone group as a compact fluorescent labeling reagent, the application of a vinyl ketone group was planned with a view to the labeling of the thiol group by the particular 1,4-addition reaction of thiol groups.

active compounds by a typical 1,4-addition reaction. In addition, the fluorescence color and intensity of **2** might be dramatically changed by the addition of a thiol group due to the change of electron state from conjugation to deconjugation. Thus, we started the synthesis of vinyl ketone derivatives of TAP.

**Synthesis of vinyl ketone analog of TAP.** The synthesis of vinyl ketone analog **2** was started with 2-methoxycarbonyl-1,3a,6a-triazapentalene **4**, which was readily obtained by the previously reported click-cyclization-aromatization cascade reaction[15]. The hydrolysis of **4** followed by condensation with *N,O*-dimethylhydroxylamine afforded Weinreb amide **5** in an 87% two-step yield. However, the treatment of **5** with vinylmagnesium bromide gave not vinyl ketone **2** but amine **6a** derived from a 1,4-addition reaction of the hydroxylamine with the resulting vinyl ketone. Although various conditions were examined, including different conditions in the work-up, the desired vinyl ketone **2** was not obtained due to the high reactivity of vinyl ketone. Similarly, the use of 1-propenyl magnesium bromide as a nucleophile afforded amine **6b**, and no vinyl ketone analogs were detected. In contrast, nucleophilic addition of isobutenyl magnesium bromide resulted in the desired vinyl ketone **2a** in good yield, because the dimethyl group at the β-position of **2a** was bulky enough to disturb the 1,4-addition of hydroxylamine and other nucleophiles. Thus, the 1,3a,6a-triazapentalene analog directly connected with vinyl ketone at the 2-position was obtained first, and we named **2a** "TAP-VK1" (Fig. 3).

**Reaction of TAP-VK1 with thiols.** Having prepared vinyl ketone **2a** (TAP-VK1), we next investigated the reactivity with thiols. Despite the bulky dimethyl group at the β-position, **2a** reacted smoothly with ethanethiol **7a**. After optimization of the reaction conditions, we found that treatment of thiol **7a** with 1.2 equiv. of

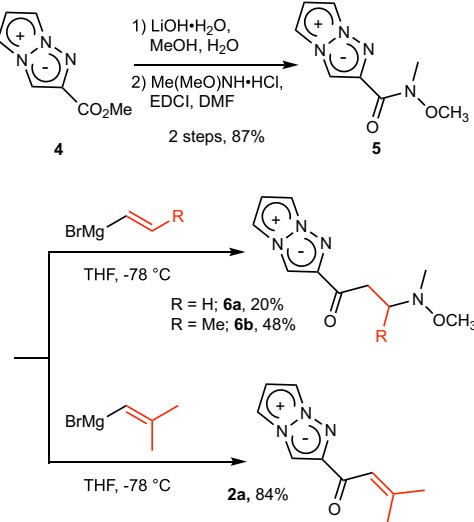

**Fig. 3 Synthesis of TAP-VK1 (2a).** 2-Methoxycarbonyl-1,3a,6a-triazapentalene **4** was successfully converted into vinyl ketone **2a** by sequential reactions consisting of hydrolysis, a condensation with the Weinreb amine, and a nucleophilic addition of isobutenyl magnesium bromide.

**2a** as a fluorescent labeling reagent in the presence of 1.0 equiv. of tetramethylguanidine (TMG) in dichloromethane (0.2 M) afforded the desired 1,4-adduct **3a** in 95% isolated yield. The reaction using 5 mol% of TMG also proceeded smoothly without a significant loss of yield, and **3a** was obtained in 87% yield (Fig. 4a). Interestingly, as we expected, the fluorescence properties of **2a** were dramatically changed after the addition of ethanethiol (**7a**). The 1,4-adduct **3a** showed a noteworthy shorter-wavelength shift of the fluorescence maximum from the 574 nm of **2a** to 521 nm, along with a change of florescence color from yellow to green. In addition, the fluorescence quantum yield ($\Phi_F$) was substantially increased up to 10 times (Fig. 4a)[26–29]. Therefore, we also investigated the time course of fluorescence in the reaction of **2a** with **7a** (Fig. 4b). The fluorescence color of the reaction mixture was changed to green within only 5 min of starting, and the fluorescence intensity subsequently increased with the time course and reached a maximum at 50 min (Fig. 4b). Therefore, TAP-VK1 (**2a**) was proven to be a useful compact thiol-labeling reagent that shows good reactivity and an increase of fluorescence after labeling, in which the background fluorescence of the unreacted **2a** would not affect the fluorescence observation. In addition, both **2a** and the thiol adduct **3a** also showed large positive fluorescence solvatochromism (Supplementary Table 2), as is the case with previously reported derivatives of 1,3a,6a-triazapentalenes. In clear contrast, other nucleophiles such as ethylamine, ethanol, and ammonium acetate did not react with **2a** at all under various conditions[30]. TAP-VK1 (**2a**) thus has great potential as a thiol specific fluorescent labeling reagent that does not need purification after labeling. The quantum chemical calculation using TD-DFT method revealed that the shift of emission maximum attributed to the decreased charge-transfer character in **3a**, where the electronic conjugation of the 2-substituents is terminated in 3a. (For detail information, see Supplementary Note 2 and Supplementary Fig. 1).

**Reaction of 2a with various thiols in dichloromethane.** Having established the optimum conditions for the thiol-labeling in organic solvent, we next examined the reactions with various thiols (Fig. 5). The reactions of aliphatic thiols such as 1-dodecanethiol **7b**, allylthiol **7c**, and benzylthiol **7d** proceeded smoothly to give the desired 1,4-adducts in 86%, 88%, and quantitative yields, respectively. Both

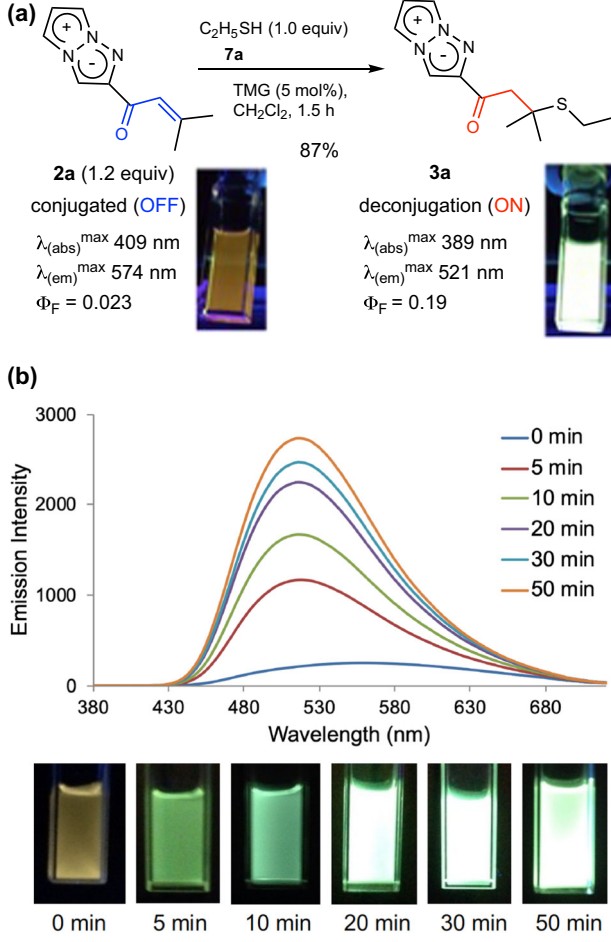

**Fig. 4 Reaction of 2a with ethanethiol (7a) and the time course of fluorescence in the reaction. a** TAP-VK1 (**2a**) smoothly reacted with ethanethiol (**7a**) to afford the more highly fluorescent **3a** in good yield, and **2a** did not react with amine and alcohol. **b** The fluorescence color of the reaction in dichloromethane changed to green from yellow after 5 min, and the fluorescence intensity increased with the time course and reached a maximum at 50 min (exited at 370 nm in $CH_2Cl_2$). TMG, 1,1,3,3-tetramethylguanidine.

thiol groups of 1,3-propanedithiol **7e** were also readily reacted with 2.2 equiv. of **2a**; i.e., the 1,4-additions of thiol to **2a** occurred twice, and the bis-TAP adduct was obtained in 86% yield. Interestingly, thiols possessing other nucleophilic functional groups, such as alcohol **7f**, amine **7g**, and triazole **7h**, specifically reacted with only thiols in excellent yield, and no byproducts derived from the addition of other nucleophiles were observed. Since the amino group of **7g** served as a base, we conducted the reaction of **7g**, which has poor solubility in dichloromethane, without TMG in water. Although the 1,4-adduct from **7g** formed a seven-membered cyclic imine (see Supplementary data), the intermolecular condensation reaction of amines with ketone moieties of **2a** and/or 1,4-adducts did not occur at all under various conditions. The 1,4-addition reactions of the secondary thiol **7i**, protected amino acid **7j**, and protected dipeptide **7k** also proceeded smoothly to afford labeled products in 82%, quantitative, and 91% yields, respectively. All of the 1,4-adducts clearly showed a noteworthy shorter wavelength shift of the emission maximum and the substantial increase of the fluorescence intensity compared to those of **2a**. Therefore, TAP-VK1 (**2a**) was demonstrated to be applicable to various aliphatic thiols with fluorescence turn-on. It is also noteworthy that only 1.2 equiv. of TAP-VK1 is enough for the labeling of thiol. In

contrast, several aromatic thiols did not react with **2a** (Supplementary Table 3).

**Reaction of 2a with peptides and protein in water.** Since the reaction of TAP-VK1 (**2a**) with **7g** proceeded smoothly in water, we suspected that **2a** has potential for use in the labeling of various biologically active compounds in water (Fig. 6). We therefore attempted to directly label a free glutathione **7l** in a water solution, and the reaction of **7l** with 2.0 equiv. of **2a** in sodium phosphate buffer solution (pH 7.0) at room temperature was found to proceed smoothly. We directly subjected the crude **3l** to reverse-phase silica gel column chromatography to give pure **3l** in 47% yield. Although the $^1$H NMR of the crude **3l** suggested that the reaction proceeded quantitatively, the isolated yield of **3l** was decreased due to the adsorption of **3l** on silica gel. Thus, we found that 0.1 M sodium phosphate buffer solution (pH 7.0) without additives was the best reaction condition for the labeling of free peptides and protein (vide infra). We next attempted to label a more complicated peptide to further assess the labeling potential of TAP-VK1 (**2a**) (Fig. 6). We synthesized octapeptide **7m** as a challenging substrate that possesses many reactive functional groups, including guanidine, alcohol, imidazole, amine, and phenol. When the octapeptide **7m** was incubated with 20 equiv. of **2a** in 0.1 M sodium phosphate buffer (pH 7.0) for 6 h, the cysteine residue of **7m** was clearly reacted to give labeled peptide **3m** in quantitative yield. The high-performance liquid chromatography (HPLC) analysis of the above reaction mixture clearly showed that there were no peaks other than the labeled product (**3m**) and excess labeling reagent (**2a**); i.e., no byproducts that reacted at other residues and no degradation products of **2a** were observed (Supplementary Fig. 2). It was thus demonstrated that **2a** specifically reacted only with a thiol group and did not react with other functional groups at all, even when a large excess amount of **2a** was used. In addition, the labeling reaction using TAP-VK1 (**2a**) was extended to bioconjugation with human serum albumin (HSA), which possesses only one free cysteine residue[31], and the HSA was cleanly labeled by **2a** (Supplementary Fig. 3). TAP-VK1 as a fluorescent labeling reagent was thus confirmed to be also applicable to proteins.

**Model imaging of TAP-VK1.** Our experiments revealed that TAP-VK1 (**2a**) is applicable to directly label peptides and protein in water. However, the above compounds **3l** and **3m** labeled by TAP-VK1 (**2a**) did not exhibit fluorescence in water, which was distinct from the case with organic solvent. In addition, **2a** itself does not emit any light in water. In contrast, the absorption intensity and the molar absorption coefficient (ε) of the labeled compounds (**3l**, **3m**) and **2a** in water were similar to or higher than those in dichloromethane. We therefore expected that these labeled compounds would also generate fluorescence under hydrophobic conditions; i.e., the labeled compounds would become luminescent when delivered to a hydrophobic area. If so, **2a** might suppress the background fluorescence from the culture medium. To investigate the fluorescence switching of **2a** between hydrophobic and hydrophilic areas, we next attempted to deliver a water-soluble compound labeled by **2a** into cells. We prepared a water-soluble peptide **7n**, including a membrane-permeable R8 peptide as a model of a bioactive compound, and we incubated it with 2.0 equiv of **2a** in 1 mM sodium phosphate buffer at room temperature (Fig. 7). After 3 h, an HPLC analysis showed that 71% of **7n** was converted into **3n** (see experimental details section in supplementary information), and we diluted the reaction mixture with Dulbecco's Modified Eagle Medium-fetal bovine serum (DMEM-FBS) (−). The resulting mixture of **3n** was directly treated with A549 cells and monitored after 18 h with excitation and emission wavelengths of 405 and 490

**Fig. 5 The reaction of 2a with various thiols.** Various thiols were smoothly labeled by **2a** in good yields, and all 1,4-adducts showed a noteworthy shorter wavelength shift of the emission maximum and a substantial increase of the fluorescence intensity. The isolated yield, the fluorescence maximum wavelength and the fluorescence quantum yield of 1,4-adducts are shown. Dithiol **7e** was treated with 2.2 equiv of **2a**, and the reaction with **7g** was conducted without base in water.

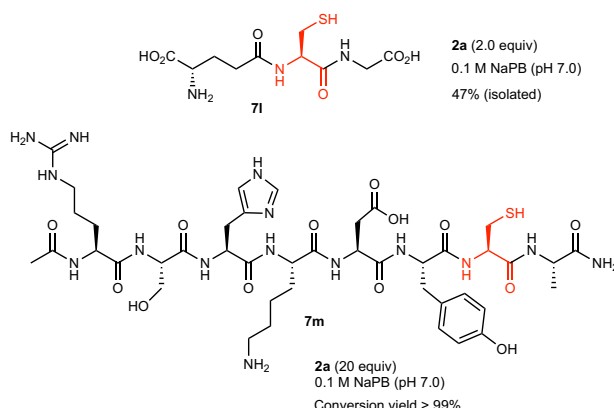

**Fig. 6 Reactions of 2a with peptides in aqueous solution.** The thiol groups of cysteine were smoothly labeled by **2a** in aqueous solution.

nm, respectively (Fig. 7). As we expected, **3n**, which does not show fluorescence in water, became luminescent within A549 cells. Treatment with a 10 μM mixture of **3n** visualized living A549 cells under fluorescence microscopy without washing of the medium culture (Fig. 7a), and the fluorescence of **3n** inside cells was further clearly observed when a 25 μM mixture was used (Fig. 7b). In clear contrast, the interiors of the control cells that were treated with a 10 μM solution and a 25 μM solution of peptide **7n** in DMEM FBS (−) and 10 μM and 25 μM solutions of TAP-VK1 (**2a**) in DMEM FBS (−) containing 0.1% DMSO were not stained (Fig. 7c–f). The fluorescence intensities of **3n** inside the cells exhibited concentration-dependence, whereas treatments with a solution of TAP-VK1 did not result in concentration-dependence (Supplementary Fig. 4). These experiments clearly indicated that the labeled peptide **3n** was incorporated through macropinocytosis of the R8 peptide[32,33], and the fluorescence of **3n** became to emit after being carried into the cells. We considered that **3n** was encapsulated by the membrane of

intracellular vesicles as the common mechanism of macro-pinocytosis so that **3n**, partially localized in the hydrophobic membrane region, recovered fluorescence. The vesicles were delivered to organelles, exhibiting fluorescence in the cytoplasm. The advantage of TAP-VK1 was that the reaction mixture of thiol-labeling could be directly used for functional analyses without the isolation and purification of the labeled substrate, because the reaction in sodium phosphate buffer (pH 7.0) did not require any reagents other than **2a**, and excess **2a** did not disturb the fluorescence observation and did not show cytotoxicity, unlike other compact fluorescent labeling reagents. Actually, when excess amount of coumarin-maleimide (**8**), 4-Chloro-7-nitro-benzofur-azan (NBD-Cl) (**9**), and 4-Fluoro-7-sulfamoylbenzofurazan (ABD-F) (**10**) as the commonly used fluorescent labeling reagents for thiols were directly treated with vascular endothelial cells (MBEC 4) as a model cells (vide infra), these fluorophores adsorbed on the cell surface and also induced the cell damage (Supplementary Fig. 5). Therefore, the contamination of excess amount of these labeling reagents obviously disturbs the fluorescence observation of living cells, so purification and isolation of labeled substrates should be required. While, TAP-VK1 (**2a**) did not affect the cells and the fluorescence observation (Supplementary Fig. 5) so that the direct treatment of the labeling reaction mixture is possible. On the other hand, treatment with a 50 μM solution of **3n** caused cell damage due to the toxicity derived from R8 peptide (Supplementary Fig. 6).

**Drug imaging with TAP-VK1.** Having confirmed the fluorescence of TAP-VK1 (**2a**) in the cells, we finally attempted to image small-molecule drug using **2a**. We adopted captopril (**7o**) as a small biologically active compound possessing thiol group. Captopril is a drug that inhibits angiotensin converting enzyme (ACE) and has been used for treatment of hypertensive patients[34]. Although the measurement of the captopril (**7o**) content in biological sample such as cells, tissues, urine and so on has been intensively studied[35–37], there is to our knowledge no report of the fluorescence imaging of **7o** inside cells. The main reason is probably that the introduction of fluorescent labeling group to **7o** induces loss of the

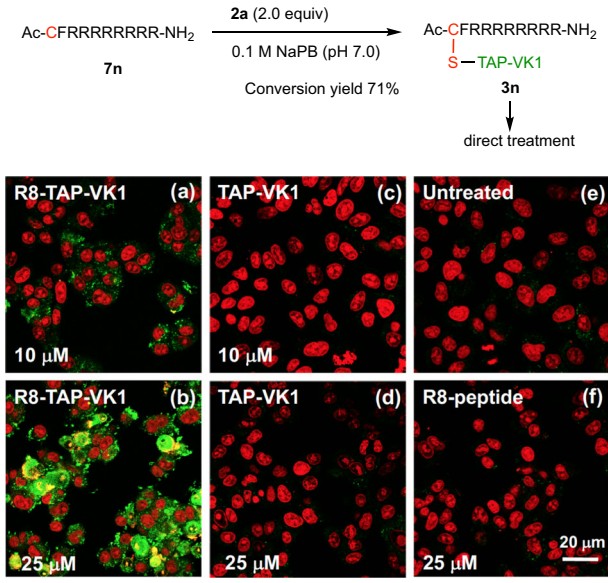

**Fig. 7 Observation of 3n in A549 cells.** Living cells were cultured with 10 μM or 25 μM of **3n** (**a**, **b**), cultured with 10 μM or 25 μM of **2a** containing 0.1% DMSO as a control (**c**, **d**), left untreated (**e**), or cultured with 25 μM of **7n** containing 0.1% DMSO (**f**). The fluorescence of **3n** inside the cells was clearly visualized (**a**, **b**), whereas that inside the cells treated by TAP-VK1 (**c**, **d**), the untreated cells (**e**) and the cells treated with R8 peptide (**f**) was not visualized under several concentrations. Cell nuclei were stained by PI (propidium iodide; red fluorescence) for detailed fluorescence observation, and **3n** was observed in the cytoplasm. The uptake of **3n** was monitored by confocal microscopy (ZEISS LSM700; Carl Zeiss) in a fluorescence image with excitation and emission wavelengths of 405 and 490 nm, respectively (**a**–**f**). Scale bar = 20 μm.

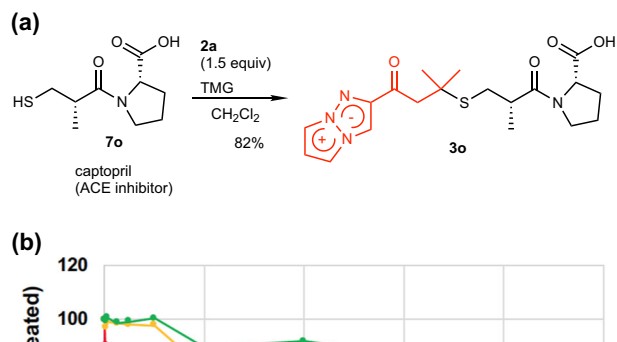

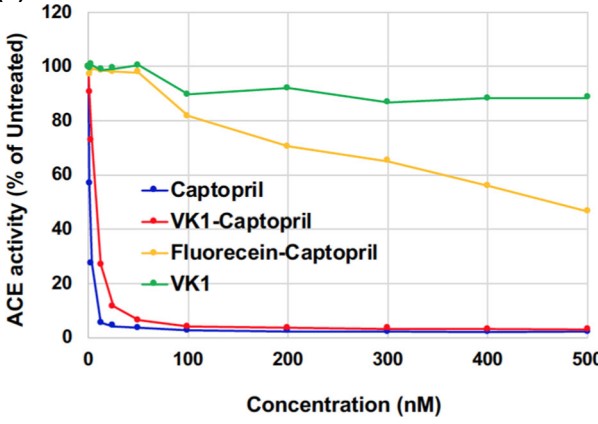

**Fig. 8 ACE activity assay of captopril and its fluorescent labeled derivatives. a** Labeling reaction of captopril (**7o**) with TAP-VK1 proceeded smoothly to give **3o** in good yield. **b** The introduction of TAP-VK1 did not affect the potent ACE inhibitory activity even with small drugs like captopril, whereas fluorescein labeling induced loss of the activity. We confirmed that the regeneration of captopril by retro-thia-Michael reaction did not occur because no peak of captopril was detected by the HPLC analysis of the assay solution of **3o**.

inhibitory activity due to the compact molecular size of captopril. Furthermore, since the probes that intracellularly react with thiols also have problems with nonspecific labeling and low labeling rate, accurate visualization of **7o** is difficult. Thus, fluorescent labeling of **7o** without loss of inhibitory activity is the most proper approach for the imaging of **7o** inside cells, and compact fluorescence labeling reagent is significant to this end. First, we examined the labeling of **7o** with TAP-VK1 and its ACE inhibitory activity. Treatment of **7o** with 1.5 equiv. of **2a** and 1.1 equiv. of TMG in dichloromethane afforded labeled captopril (**3o**) in 82% yield as a 4:1 diastereomeric mixture at the methyl group (Fig. 8). Since the epimer of **7o** is known to be inactive[34,38], the diastereomeric mixture of **7o** was applied for the ACE activity assay using ACE-Kit WST (See the "Methods" section). Surprisingly, TAP-VK1-labeled **3o** retain the potent inhibitory activity of captopril, whereas fluorescein-labeled captopril dramatically lost the activity (Fig. 8). Thus, it was found that the introduction of TAP-VK1 does not affect the activity even for very small molecules such as **7o**, and it clearly proved that TAP-VK1 is powerful tool for the fluorescence labeling of small biologically active compound such as drugs, natural products, hormones, peptides, and so on.

Having confirmed the ACE inhibitory activity of captopril (**7o**) labeled by TAP-VK1, we finally attempted to visualize the localization of captopril inside vascular endothelial cells. A 50 μM solution of **3o** in DMEM FBS (−) was treated with mouse brain-derived vascular endothelial cells (MBEC4) at 37 °C, and the cells were monitored by confocal laser microscope with excitation and emission wavelengths of 405 and 490 nm, respectively, at 24 h after treatment (Fig. 9). As we expected, the observable fluorescence appeared in immobilized cells after 24 h (Fig. 9a, b), whereas no fluorescence was observed in the untreated control cells

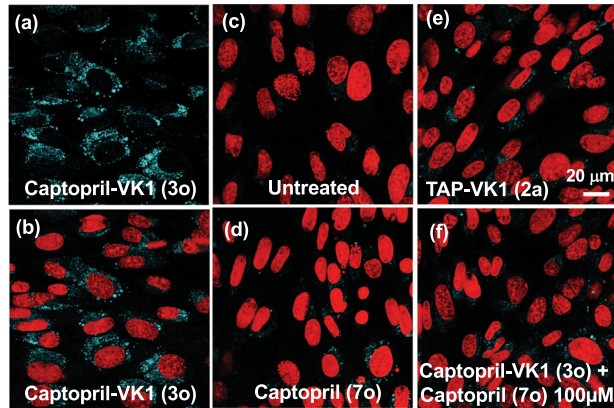

**Fig. 9 Observation of captopril in vascular endothelial cells.** Living cells were cultured with 50 μM of **3o** (**a**, **b**), left untreated (**c**), cultured with 50 μM of **7o** and 50 μM of **2a** containing 0.1% DMSO as controls (**d**, **e**), or cultured with the mixture of 50 μM of **3o** and 100 μM of **7o** as a competition experiment (**f**). The fluorescence of **3o** inside the cells was clearly visualized (**a**, **b**), whereas no fluorescence was observed inside of the untreated cells (**c**). Inside of cells treated by captopril (**d**) and TAP-VK1 (**e**) showed very weak fluorescence, and their fluorescence intensities were much lower than that of **3o**. The presence of 2.0 equiv. of captopril competed with **3o** and substantially reduced the fluorescence of **3o** (**f**). The Cell nuclei were stained by PI (propidium iodide; red fluorescence) for detailed fluorescence observation (**b**–**f**). The fluorescence of **3o** was observed in the cytoplasm. The uptake of **3o** was monitored by confocal microscopy (ZEISS LSM700; Carl Zeiss) in a fluorescence image with excitation and emission wavelengths of 405 and 490 nm, respectively (**a**–**f**). Scale bar = 20 μm.

(Fig. 9c). On the other hand, the interiors of the cells that were treated with a 50 μM solution of captopril **7o** in DMEM FBS (−) and 50 μM solutions of TAP-VK1 (**2a**) in DMEM FBS (−) showed weak fluorescence (Fig. 9d, e), however, the fluorescence intensities were significantly lower than **3o** (Supplementary Fig. 7). In addition, competition experiments with TAP-VK1-labeled captopril **3o** and unlabeled captopril **7o** reduced the intracellular fluorescence (Fig. 9f), suggesting that TAP-VK1-labeled captopril **3o** accumulated in ACE and resulted in fluorescence staining of ACE.

In addition, to evaluate the advantages of TAP-VK1 (**2a**) as the novel compact fluorophore (5 + 5 membered ring), commonly used small molecular fluorophores such as coumarin (**8**) (6 + 6 membered ring), NBD (**9**) (6 + 5 membered ring), and ABD (**10**) (6 + 5 membered ring) were similarly introduced into the thiol group of captopril to give captopril-coumarin (**8o**), captopril-NBD (**9o**), and captopril-ABD (**10o**), respectively (Supplementary Table 4). Similar treatment of these labeled captoprils with MBEC4 revealed that **2a** was the most efficient fluorophore that retained the original behavior of captopril (Supplementary Fig. 8). The fluorescence of **9o** and **10o** were not observed inside the cells, i.e., NBD and ABD probably disturbed the delivery of captopril into cells. On the other hand, although **8o** was incorporated into the cells and emitted fluorescence inside cells, the fluorescence intensity was much lower than that of Captopril-VK1 (**3o**), and the distribution of **8o** appeared to be more non-specifically localized compared to that of **3o** (Supplementary Fig. 8).

Finally, to confirm that **3o** fluorescently stained ACE, a comparative experiment with antibody staining was performed (Fig. 10a–d). The fluorescence distribution of antibody staining experiment of ACE in MBEC 4 cells was co-localized with the fluorescence of TAP-VK1-labeled captopril **3o**. The image of antibody staining in Fig. 10 is similar to the previously reported immunolocalization of ACE in CHO and HUVEC cells[39,40], and our imaging properly showed the behavior of ACE in cultured vascular endothelial cells. Thus, by using TAP-VK1 (**2a**), we first succeeded in the visualization of the captopril (**7o**) binding to the target protein in vascular endothelial cells. It is noteworthy that very small drug such as captopril can be fluorescently labeled without loss of biological activity. On the other hand, the coumarin analog **8o** were partially distributed to ACE, but were also localized in other area (Fig. 10e–h), while TAP-VK1 analog **3o** mainly co-localized in ACE. Therefore, TAP-VK1 is considered to be more effective fluorophore in tracking the intracellular localization of the target compound. Furthermore, to evaluate the photostability of **3o**, the MBEC 4 cells treated with **3o** and **8o** were exposed to the continuous excitation laser at 405 nm (5 mW Diode Laser) and measured their fluorescence intensity. The fluorescence half-life time ($T_{1/2}$) of **3o** (76.9 min) was longer than that of **8o** (41.5 min), and TAP-VK1 was found to possess good photostability (Supplementary Fig. 9).

## Discussion

Fluorescence imaging is one of the most powerful approach for the mechanistic study of biologically active compound. However, fluorescence imaging of biologically active small compounds in cells and tissues is generally difficult because the introduction of fluorescent labeling groups induces the change of biological activities of parent small compound according to the molecular

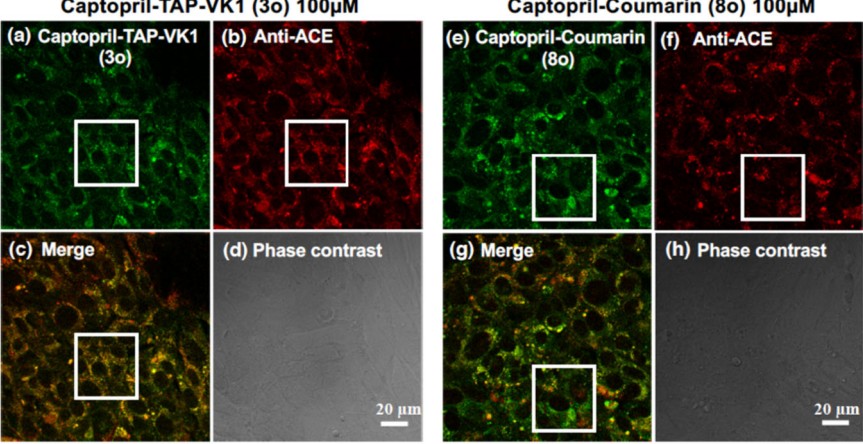

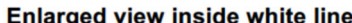

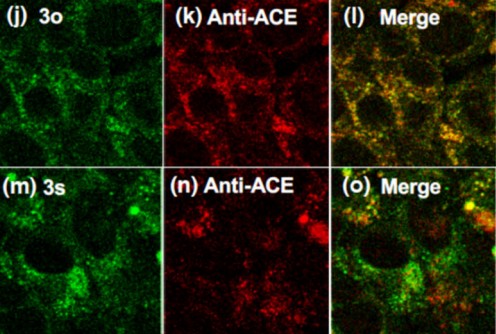

**Fig. 10 Observation of captopril derivatives (3o and 8o) in vascular endothelial cells.** Living cells were cultured with 100 μM of **3o** and **8o**. The fluorescence distribution of **3o** was mainly consistent with that of ACE fluorescently stained with anti-ACE antibody (**a**–**d**), while the fluorescence distribution of **8o** showed incomplete consistent compared to **3o** (**e**–**h**). The enlarged images inside white line show the difference more clearly (**j**–**o**). The uptake of **3o** and **8o** were monitored by confocal microscopy (ZEISS LSM700; Carl Zeiss) in a fluorescence image with excitation and emission wavelengths of 405 and 490 nm, respectively. Scale bar = 20 μm.

size and/or physical properties of fluorescent labeling groups. To solve this problem, we herein developed a compact and thiol-specific fluorescent labeling reagent, TAP-VK1 (**2a**).

TAP-VK1 (**2a**) smoothly react with various aliphatic thiols in organic and water solvent to give 1,4-adducts **3** in high yield, and other nucleophiles such as alcohols, amines, and carboxylate did not react with **2a** at all. The validity of **2a** as fluorescent labeling group was confirmed by TAP-VK1-labeled R8 peptide that permeates into cells and emit fluorescence inside cells. Indeed, the utility of **2a** as the compact labeling reagent was demonstrated by the labeling of captopril, i.e., TAP-VK1-captopril retains potent ACE inhibitory activity, and its fluorescence accumulated in ACE in vascular endothelial cells.

TAP-VK1 (**2a**) has various advantages in addition to the compact molecular size as follows: (i) high thiol specificity, (ii) convenient reaction condition for labeling, (iii) high labeling yield in both organic solvents and water, (iv) significant increase in fluorescence intensity after labeling so that background fluorescence of unreacted **2a** do not disturb fluorescence observation, while the commonly used compact labeling reagents disturbed, and (v) direct treatment of the labeling reaction mixture is possible. On the other hand, there are still some points to be improved in **2a** such as increase of intracellular fluorescence intensity. To this end, tuning the **2a** by introducing hydrophobic substituents so as to emit fluorescence in water is required, we will undertake this study in the future. Although improvement of **2a** is still necessary, the first successful imaging of captopril, one of the most compact drugs, in this study demonstrated the usefulness of compact fluorescent probes for mechanistic study. Thus, development of compact molecular probes will be an intensive research topic in the future.

## Methods

**Chemical synthesis**. Detailed synthetic procedures and compound characterization are described in the Supplementary Methods, and $^1H$ and $^{13}C$ NMR and absorption and fluorescence emission spectra are also available (Supplementary Figs. 14–57).

**Human serum albumin for bioconjugation**. Human serum albumin (Kenketsu Albumin 20 "KAKETSUKEN") was purchased from KAKETSUKEN (Kumamoto, Japan). The albumin was defatted by treatment with charcoal as described in literature[41], dialyzed against deionized water, lyophilized, and stored at –20 °C until used. To a solution of HSA (20.4 μM in 0.1 M sodium phosphate, pH 7.0, 98 μL) was added TAP-VK1 (**2a**) (100 mM in DMF, 2 μL) and the mixture was incubated at room temperature for 6 h. Then 5.0 μL aliquot was treated with 8.7 μL of sample buffer (50 mM Tris-HCl, 2.0% (v/v) SDS, 6.0% (v/v) 2-mercaptoethanol, 10% (v/v) glycerol and 0.050% (w/v) bromophenol blue in $H_2O$) and heated at 100 °C for 2 min.

**Cell lines and culture**. A549 cell line (obtained from RIKEN Cell Bank) and MBEC4 cells[42] were grown in Dulbecco's Modified Eagle's Medium (DMEM: Sigma) containing 10% fetal bovine serum (FBS: Life Technologies), 70 mg/L penicillin (Sigma) and 100 mg/L streptomycin (Sigma), and maintained in the same culture medium at 37 °C in 5% $CO_2$.

**Confocal imaging**. A549 cells or MBEC4 cells ($5 \times 10^4$) were plated on 8well-chamber slide (Nunc) and incubated with DMEM containing 10% FBS for a day. The medium was replaced with serum-free DMEM containing compounds. The cells were fixed with 4% paraformaldehyde (PFA) after incubation for 24 h at 4 °C. Fixed cells were washed with phosphate based saline (PBS), and then stained with 2 μg/mL propidium iodide (Sigma) containing 20 μg/mL RNase A (Sigma) for 1 h at room temperature. The specimens were viewed under a confocal fluorescent microscopy (LSM700, Zeiss). After treatment of compounds, the MBEC4 cells were fixed with 4% PFA for 24 h at 4 °C, and then immunostained by a two-step incubation method. Fixed cells were treated at 4 °C overnight with the appropriately diluted anti-ACE antibody (Santa cruz). After washing with PBS containing 0.05% Tween20, the cells were treated with Alexa Fluor 555-conjugated anti-mouse IgG (Cell Signaling Technology). The specimens were viewed under a confocal fluorescent microscopy.

**Measurement of cellular fluorescence intensity**. Digital imaging analysis was also performed by a standard imaging software attached to the confocal microscopic system in a fluorescence mode. The cells containing significant fluorescent inclusions were designated as positive cells. Three different areas showing 20–30 cells each were viewed. The relative amount of the accumulated natural substrates in microglial cell population was expressed as immunofluorescence intensity index in each image (mean value). Relative fluorescence intensity was graded, ranging from 0 to 255 in this system. The fluorescence intensity was linearly correlated with the number of pixels showing a positive signal. Relative intensities for other cell strains were calculated within the range of this gradation.

**Computational details**. The equilibrium geometry in the electronic ground ($S_0$) is determined by the density functional theory (DFT) calculations using the ωB97XD functional[43] with the 6-311+G(d,p) basis set, while the geometry optimization in the singlet excited state ($S_1$) is performed by the time-dependent DFT (TD-DFT) calculations employing the same functional and basis set. In geometry optimization, no symmetry constraint is imposed, and the solvent effects (dichloromethane) are taken into account by the polarizable continuum solvation model (PCM). After the geometry optimizations, the vertical excitation and emission energies are calculated at the $S_0$ and $S_1$ equilibrium structures (denoted as $(S_0)_{min}$ and $(S_1)_{min}$), respectively. In calculating the excitation energies at $(S_0)_{min}$ and $(S_1)_{min}$ in dichloromethane, the solvent effects are taken into account by the vertical excitation/emission model (VEM)[44] implemented in the VEMGAUSS package. All DFT calculations are performed using the Gaussian09 program package[45].

**Angiotensin converting enzyme activity assay**. The inhibition of ACE activity according to captopril and its fluorescent labeled derivatives was performed using Dojindo ACE Kit-WST test kit (Dojindo Laboratories, Kamimashiki-gun, Kumamoto. Briefly, the enzymatic reaction was initiated by the ACE and aminoacylase in the mixture containing 3HB-GGG (3-hydroxybutyrate glycylglycylglycine) and the ACE inhibitor. The mixture was incubated at 37 °C for 60 min. During this incubation, the substrate, 3HB-GGG, was enzymatically cut into 3HB-G and G-G and then 3HB and G. The yield of 3HB was monitored indirectly through formazan concentration, which was measured at 450 nm after 10 min at 25 °C. Absorbance was measured at 450 nm using the microplate reader INFINITE F NANO Plus (TECAN).

**Measurement of absorbance/fluorescence spectra and fluorescence quantum yields**. Absorption spectra were recorded on a JASCO V-600 spectrometer and corrected fluorescence spectra were recorded on a JASCO FP-8200 spectrofluorometer. Sample solutions were degassed thoroughly by purging with an Ar gas stream for 30 min prior to the experiments and then sealed in their cells. All measurements were carried out at 25 °C. Fluorescence quantum yields were estimated by using 9,10-diphenylanthracene (9,10-DPA) in cyclohexane ($\Phi_F = 0.91$) or rhodamine B in ethanol ($\Phi_F = 0.94$) as a standard. Values were calculated by using the Eq. 1.

$$\Phi_x = \Phi_{st}[A_{std}/A_x][F_x/F_{std}]\left[n_x^2/n_{std}^2\right] \qquad (1)$$

std: standard, x: sample, A: absorbance at the excitation wavelength, F: fluorescence spectrum area, n: refractive index

## Data availability
The main data supporting the finding of this study are available within the paper and its Supplementary Information file. Other relevant data are available from the corresponding author upon reasonable request.

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

## Acknowledgements

This work was partially supported by JSPS KAKENHI Grant Numbers JP16H01156 in Middle Molecular Strategy, JP18H04416, JP17K008365, and JP19H02851. We acknowledge Tokushima University for their financial support of the Research Clusters program of Tokushima University (No. 1802001).

## Author contributions

K.N. conceived the experiments and analyzed the results. Atsushi Nakayama, A. Otani and H.M. performed the laboratory experiments and optimized the reaction conditions. T.I. and A. Otaka performed the synthesis of peptides. D.T. and K.I. performed the fluorescence imagings and analyzed the results of cell staining experiments. A. Nakayama performed the DFT calculations for the fluorescence mechanism of TAP-VK1. K.N. wrote the paper, and K.T. proofread the paper.

## Competing interests

The authors declare no competing interests.
