## [Peer Review File · Communications Chemistry]

Reviewers' comments:

Reviewer #1 (Remarks to the Author):

This is a report about the application of a new enone derivative of a weakly fluorescent triazapentalene, that can its fluorescence have turned on with thiols after 1,4-addition. There is at the same a hypsochromic shift. The work has its merit and has been carefully done. The authors have published previously on this relatively new chromophore. The concept of turning on the fluorescence in this way by thiol addition has been done with other chromophores but is new for triazapentalenes.

A small remark about the need for small grammatical corrections, for instance in the nonconsequent use of the singular/plural (as we can already see in the title).

Correct also "fluorescein" in Figure 8

Otherwise it can essentially be published as such.

Reviewer #2 (Remarks to the Author):

In this manuscript, Namba and coworkers reported the developing of special triazapentalene building block (TAP-VK1) as thiol-specific fluorescent labeling reagent and its application in solution and live cell system. This triazapentalene molecule was developed based on authors previous works (ref 15, 25). The key discovery here is the incorporation of enone group on TAP. Upon thiol Michael addition, significantly increasing of FL emission was recorded. There are some unique features of this reaction, including the ability to specifically target various aliphatic thiol groups without touching other nucleophiles existing in biosystem. The reaction can also take place smoothly both in organic and water solvent system. Peptide and enzyme modified probe has been successfully applied in cell imaging. Overall, this work was conducted with high quality. However, this reviewer has some concerns regarding the novelty and practical useful issues:

1) This system relies on the thiol Michael addition to form the actual FL active molecule. The actual probe is TAP-VK1-Peptide. Thus, by strictly definition, this is not an actual turn-on sensor. It just like a peptide probe with fluorescence tag. Therefore, it is unclear what is the unique advantage of this FL-tag modified peptide comparing with other FL-tag.

2) If the function is simply a thiol-based FL-tag, it is important to evaluate its photo stability and other related property. Are there any photo-bleach occurring over time? How stable of the FL-tag over time etc?

It is important to note that there are a lot of beautiful experimental design in this work and the cell line studies shown in figures 7-10 are impressive. However, considering that no clear justification has been made why this TAP-VK is clearly better than existing systems and full photo property characterization of this new dye, this reviewer do not fell comfortable to recommend the work for publication to high impact journal like Nature Communication. It will fit in perfectly with journals targeting more specific audiences.

Reviewer #3 (Remarks to the Author):

Manuscript by Nakayama A. et al. describes the synthesis and applications of thiol-modifying fluorescent tag based on small molecular weight fluorophore 1,3a,6a-triazapentalene (TAP). Based on the fluorophore scaffold originally developed by the authors' group, they extended the design with removing the aromatic group and attaching the optimized a,b-unsaturated ketone structure to develop a small sized thiol-modifying reagent. The molecule was able to selectively label thiols in small molecules and peptides. The thiol adducts were not fluorescent in aqueous media, but became fluorescent in non-polar solvent, and authors were able to visually analyze the

cellular distributions of labeled molecules by fluorescence imaging.

I think that chemical and photochemical characterizations of the system were solidly performed throughout the manuscript. Considering the potential usefulness of the molecule in fluorescence-based bioanalysis, I think that the manuscript is potentially publishable in *Commun. Chem.*. However, I suggest that the advantage of their tag molecule over other present molecules should be discussed more thoroughly with additional data.

Firstly, I agree that connected five-membered aromatics would be one of the smallest chemical structures to emit fluorescence at visible light region, but there are other small molecular fluorophores such as coumarins (6 + 6 membered ring) and NBD (6 + 5 membered ring) that are commonly used to label biomolecules. Authors should discuss about the conspicuous advantages of their fluorophores (e.g. absorbance/fluorescence wavelengths, brightness, chemical and photochemical stabilities) compared to them.

In relation to this, the criteria of the good tag molecule are not only in their molecular sizes. I consider that it is also important that the tag exhibits the minimal effects on their physical properties (e.g. hydrophobicity, charges) of the original compound, so the modification does not significantly alter the cellular distribution and does not induce non-specific interactions with other proteins. Authors showed that the derivatization of captopril did not lose the activity toward ACE, but they should test how the hydrophobicity (e.g. water-octanol distribution) and non-specific protein binding (e.g. plasma protein binding) were affected by the modification.

In addition, I suggest that authors should characterize and discuss the mechanism of fluorescence enhancement of labeled molecules in living cells more solidly, so we can properly evaluate the results.

In case of R8-peptide, authors vaguely discussed about it using the term "hydrophobic area", but what exactly does it mean? Does it mean the inner membrane compartment, or is the high protein concentration in cytosol (around 200 mg/mL) sufficient to induce the non-specific interaction with the peptide to cause fluorescence?

In case of captopril derivative, it is highly important to characterize the system in *in vitro* experiment using recombinant proteins. Does fluorescence enhancement occur by drug's binding with ACE? And does it have sufficient selectivity against other non-specific proteins?

Minor points

- In modification of HSA, "HSA possesses only one free cysteine residue", needs a reference.
- Methodological details in the measurement of absorbance/fluorescence spectra and quantum yields should be given. In Figure 4 (B), excitation wavelength should be shown. In "Absorption and Fluorescence emission spectra" section in SI, excitation and emission wavelengths should be shown.
- In Figure S6, the number of experiments (n) and the meaning of the error bar (S. D. or S. E.) should be shown.

Re: Revision of our Manuscript COMMSCHEM-19-0225

We deeply appreciate reviewers' kind suggestions and corrections for making our manuscript more valuable. We have revised the manuscript according to the suggestions of reviewers 2 and 3 as explained below. Reviewer 1 required no changes. Also, we attached the 'track changes' version, which was highlighted the revised and rewrote parts, for easy review.

All the files required were sent from manuscript tracking system following your guide, so that they will soon be in your hand.

Sincerely yours,

Our responses to the suggestion of Reviewer 2:

- 1) We showed the unique advantages of TAP-VK1 as a fluorescent labeling reagent comparing with other commonly used small fluorescent labeling reagents by the additional experiments (Supplementary Table 4 and Fig. 5). That is, while other reagents disturbed the fluorescence observation of living cell due to their cytotoxicity and adsorption on the surface of cells, excess TAP-VK1 did not. Thus, the mixture of labeling reaction can directly use for the fluorescence observation.
- 2) Photostability of TAP-VK1 inside the cell was evaluated (Supplementary Fig. 9). The TAP-VK1-labeled captopril showed longer half-life time (76.9 min) than that of coumarin-labeled captopril (41.5 min), and the good photostability of TAP-VK1 was demonstrated. Moreover, we clearly demonstrated the advantages of TAP-VK1 compared with existing small fluorophores for the fluorescence imaging of biologically active compound in the cells by the additional experiments (Fig.10, Supplementary Figs. 5, 8, and 9). Therefore, we now believe that the clear justification has been made for the merits of TAP-VK1 as the novel fluorescent labeling reagent.

Our responses to the suggestion of Reviewer 3:

[For the main criticism of Reviewer 3]:

We elucidated the advantages of TAP-VK1 as a novel fluorescent labeling reagent compared with the commonly used small fluorescent labeling reagents by additional experiments (Fig. 10, Supplementary Table 4, Figs. 5, 8, and 9). According to the suggestion of reviewer 3, we synthesized and evaluated the captopril analogs labeled by other existing small fluorophore such as coumarins (6 + 6) and NBD (5 + 6). Also, as NBD did not show the fluorescence, ADB (5 + 6) analog was also prepared for the fluorescence imaging in the cells. The fluorescent imaging of these labeled analogs of captopril in the mouse brain-derived vascular endothelial cells (MBEC 4) revealed that TAP-VK1 is more effective fluorophore in tracking the intracellular localization of the target compound and showed minimal effects on the behavior of original bioactive molecules.

In addition, we added our consideration about the mechanism of fluorescence enhancement of labeled R8-peptide in living cells.

[Minor points from Reviewer 3]:

1. We added reference to reference 31 that HSA possesses only one free cysteine residue. Thus, the reference numbers after 31 were shifted.
2. Methodological details in the measurement of absorbance/fluorescence spectra and quantum yield were given in Method section. Excitation wavelength in Figure 4 (B) was shown. Excitation and emission wavelengths were shown in spectra in SI section.
3. In previous figure S6 (currently S7), the number of experiments (n) and meaning of the error bar was shown.

We really appreciate the suggestions of reviewers, which solidify our research. We hope that our revision reach the level required by reviewers.

List of Additional Experiments for Revision:

1. **Supplementary Table 4**, Comparison of the fluorescence properties of Captopril derivatives labeled by other commonly used small fluorophore.
2. **Supplementary Fig. 5**, Fluorescence and phase contrast images of high concentration treatment of TAP-VK1, Coumarin-maleimide, NBD-Cl, and ABD-F to MBEC4 cells.
3. **Supplementary Fig. 8**, Fluorescence and phase contrast images of high concentration treatment of Captopril-TAP-VK1, Captopril-Coumarin, Captopril-NBD, and Captopril-ABD.
4. **Supplementary Fig. 10**, Photostability evaluation of Captopril-TAP-VK1 and Captopril-Coumarin.
5. **Fig. 10 in main text**, Comparison of fluorescence distribution of Captopril-TAP-VK1 and Captopril-Coumarin with the distribution of ACE that was fluorescently stained by anti-ACE antibody.

Other revised points according to the manuscript checklist.

1. Short descriptions of background and paper's conclusion were added to Abstract.
2. A brief summary of both the results and the conclusions were added to the end of Introduction.
3. "Data availability" was added.
4. Author contributions statement is provided.
5. Competing interest statement is provided.

REVIEWERS' COMMENTS:

Reviewer #2 (Remarks to the Author):

In this revised manuscript, the author addressed this reviewer's previously concern about the advantage of TAP-VK1 compared to other small fluorescent labeling reagents. although some concern remains, it is impressive to see the lower cytotoxicity, higher adsorption, and relative longer half-time of this new class of molecular probe. With this result and further explanation of the mechanism in the living cells, this reviewer feels it is worth to report this novel thiol-activated fluorescent labeling reagent (TAP-VK1) in Nature Communication for the interest of the community. Therefore, this reviewer will like to recommend publication of this work as is.

Reviewer #3 (Remarks to the Author):

I consider that the authors adequately revised the manuscript, and that the current manuscript serves as the important report of the novel fluorescent labeling reagent. I agree with the acceptance of the manuscript without further changes.

Re: Revision of our Manuscript COMMSCHEM-19-0225A

Dear referees,

We deeply appreciate reviewers' careful proofreading and appropriate suggestions. Referee 1 required no change in the original version, and referees 2 and 3 accepted revised version without change. Therefore, there is no response to referees in this final version.

Sincerely yours,

Reviewer #1 (Remarks to the Author in original submission):

This is a report about the application of a new enone derivative of a weakly fluorescent triazapentalene, that can its fluorescence have turned on with thiols after 1,4-addition. There is at the same a hypsochromic shift. The work has its merit and has been carefully done. The authors have published previously on this relatively new chromophore. The concept of turning on the fluorescence in this way by thiol addition has been done with other chromophores but is new for triazapentalenes.

A small remark about the need for small grammatical corrections, for instance in the nonconsequent use of the singular/plural (as we can already see in the title).

Correct also "fluorescein" in Figure 8

Otherwise it can essentially be published as such.

Reviewer #2 (Remarks to the Author in revised version):

In this revised manuscript, the author addressed this reviewer's previously concern about the advantage of TAP-VK1 compared to other small fluorescent labeling reagents. although some concern remains, it is impressive to see the lower cytotoxicity, higher adsorption, and relative longer half-time of this new class of molecular probe. With this result and further explanation of the mechanism in the living cells, this reviewer feels it is worth to report this novel thiol-activated fluorescent labeling reagent (TAP-VK1) in Nature Communication for the interest of the community. Therefore, this reviewer will like to recommend publication of this work as is.

Reviewer #3 (Remarks to the Author in revised version):

I consider that the authors adequately revised the manuscript, and that the current manuscript serves as the important report of the novel fluorescent labeling reagent. I agree with the acceptance of the manuscript without further changes.